# Adjuvant Properties of Caffeic Acid in Cancer Treatment

**DOI:** 10.3390/ijms25147631

**Published:** 2024-07-11

**Authors:** Nicole Cortez, Cecilia Villegas, Viviana Burgos, Jaime R. Cabrera-Pardo, Leandro Ortiz, Iván González-Chavarría, Vaderament-A. Nchiozem-Ngnitedem, Cristian Paz

**Affiliations:** 1Laboratory of Natural Products & Drug Discovery, Center CEBIM, Department of Basic Sciences, Faculty of Medicine, Universidad de La Frontera, Temuco 4780000, Chile; n.cortezsalvo01@gmail.com (N.C.); c.villegas04@ufromail.cl (C.V.); 2Departamento de Ciencias Biológicas y Químicas, Facultad de Recursos Naturales, Universidad Católica de Temuco, Rudecindo Ortega, Temuco 4780000, Chile; vburgos@uct.cl; 3Laboratorio de Química Aplicada y Sustentable (LabQAS), Departamento de Química, Facultad de Ciencias, Universidad del Bío-Bío, Concepción 4081112, Chile; jacabrera@ubiobio.cl; 4Instituto de Ciencias Químicas, Facultad de Ciencias, Universidad Austral de Chile, Valdivia 5110566, Chile; leandro.ortiz@uach.cl; 5Departamento de Fisiopatología, Facultad de Ciencias Biológicas Universidad de Concepción, Concepción 4030000, Chile; ivangonzalez@udec.cl; 6Institut für Chemie, Universität Potsdam, D-14476 Potsdam, Germany; nchiozem-ngnitedem@uni-potsdam.de

**Keywords:** caffeic acid, vegetal sources, metabolism, adjuvant cancer treatment

## Abstract

Caffeic acid (CA) is a polyphenol belonging to the phenylpropanoid family, commonly found in plants and vegetables. It was first identified by Hlasiwetz in 1867 as a breakdown product of caffetannic acid. CA is biosynthesized from the amino acids tyrosine or phenylalanine through specific enzyme-catalyzed reactions. Extensive research since its discovery has revealed various health benefits associated with CA, including its antioxidant, anti-inflammatory, and anticancer properties. These effects are attributed to its ability to modulate several pathways, such as inhibiting NFkB, STAT3, and ERK1/2, thereby reducing inflammatory responses, and activating the Nrf2/ARE pathway to enhance antioxidant cell defenses. The consumption of CA has been linked to a reduced risk of certain cancers, mitigation of chemotherapy and radiotherapy-induced toxicity, and reversal of resistance to first-line chemotherapeutic agents. This suggests that CA could serve as a useful adjunct in cancer treatment. Studies have shown CA to be generally safe, with few adverse effects (such as back pain and headaches) reported. This review collates the latest information from Google Scholar, PubMed, the Phenol-Explorer database, and ClinicalTrials.gov, incorporating a total of 154 articles, to underscore the potential of CA in cancer prevention and overcoming chemoresistance.

## 1. Introduction

Caffeic acid (CA) was first identified by Hlasiwetz in 1867 during the hydrolysis of caffetannic acid using caustic potash [1]. CA is a naturally occurring polyphenol found widely in the plant kingdom, particularly in beverages like coffee and yerba mate, which are consumed worldwide. Polyphenols are characterized by a benzene structure functionalized with one or more hydroxyl groups. Presently, there are over 8000 known polyphenolic compounds, categorized into various families, Figure 1, including phenolic acids (such as hydroxybenzoic acids and hydroxycinnamic acids), flavonoids (encompassing flavones, flavonols, flavonones, isoflavones, flavononols, anthocyanins, flavan-3-ols, and chalcones), tannins (both hydrolysable and non-hydrolysable (condensed) tannins), stilbenes, lignans, quinones (benzoquinone, naphthoquinone, and anthraquinones), and coumarins (simple coumarins, furanocoumarins, pyranocoumarins, benzocoumarins, coumestans, and biscoumarins) [2,3,4,5,6,7,8,9,10].

The properties of CA stem from its distinctive chemical structure. The presence of phenolic hydroxyl groups in the catechol moiety, coupled with a double bond in the carbon chain, imparts both antioxidant and pro-oxidant characteristics to CA. Within cells, CA serves as a primary antioxidant by counteracting harmful reactive oxygen species (ROS) that pose threats to deoxyribonucleic acid (DNA), proteins, and lipids, thus mitigating the risk of carcinogenesis. Moreover, CA exhibits secondary antioxidant activity by stimulating the intracellular nuclear erythroid 2-related factor 2/antioxidant response element (Nrf2/ARE) pathway. The Nrf2 regulates the expression of phase II antioxidant enzymes, including glutathione S-transferase (GST), heme oxygenase 1 (HO-1), and NADPH Quinone Dehydrogenase 1 (NQO1), which play pivotal roles in preserving cellular redox homeostasis [11,12]. Research has shown that CA exhibits hepatoprotective properties in HepG2 cells, shielding them from oxidative stress triggered by tert-butyl hydroperoxide (t-BHP). This suggests that the activation of Nrf2 and a rise in levels of HO-1 and glutamate-cysteine ligase (GCL) within the cell nuclei occur in response [13]. Furthermore, CA has been shown to mitigate cell and tissue damage in the brains of rats exposed to neurotoxic compounds such as quinolinic acid (QUIN, 100 μM), ferrous sulfate (FeSO_4_, 25 μM), and 6-hydroxydopamine (6-OHDA, 100 μM). At a concentration of 100 μM, CA reduced oxidative damage and improved brain function in rats. Similar findings were observed in the worm model *C. elegans*, where CA directly activated the Nrf2/ARE pathway [14]. Additionally, CA has been found to down-regulate inflammatory interleukins, specifically interleukin-6 (IL-6) and interleukin-1β (IL-1β), as well as nuclear factor kappa B (NF-κB) in the inflammatory cascade and inhibit signal transducer and activator of transcription 3 (STAT3) and signal-regulated kinase 1/2 (ERK1/2) actions in vitro [15,16,17]. However, an antioxidant agent such as CA can become a pro-oxidant due to its ability to chelate metals such as copper (Cu), inducing lipid peroxidation and causing DNA damage through oxidation or the formation of covalent adducts with DNA [18].

These actions play a vital role in both cancer prevention and treatment. Furthermore, evidence indicates that caffeic acid might enhance the susceptibility of cancer cells to certain drugs frequently employed in chemotherapy, implying its potential as a promising compound for combating chemoresistance. This review aims to underscore the phytochemical characteristics of CA and its direct involvement in cellular pathways linked to cancer development and advancement, in contrast to other reviews that demonstrate the general activity of CA. Here, we provide a strong basis for the use of CA as an adjuvant in cancer chemotherapy and radiotherapy.

## 2. Methodology

The literature review was conducted across multiple databases, including Google Scholar, PubMed, and Springer (Figure 2). Initially, a search was conducted using keywords such as caffeic acid, chemoresistance, adjuvant, biosynthesis, and metabolism. Subsequently, additional terms like inflammation and molecular targets were combined with caffeic acid for a more comprehensive search. The literature exploration involved examining the bibliographies of selected publications featuring original research to compile this review article.

## 3. Biosynthesis of Caffeic Acid

CA is synthesized via the phenylpropanoid pathway, which originates from the amino acids phenylalanine or tyrosine. Tyrosine undergoes a two-step conversion process to produce *p*-coumaric acid, initially catalyzed by the enzyme tyrosine ammonia-lyase (TAL). Subsequently, *p*-coumaric acid is transformed into CA by 4-coumarate 3-hydroxylase (C3H), which introduces a hydroxyl group at position 3. Another route involves the conversion of *p*-coumaric acid to coumaroyl-CoA by 4-coumarate (i.e., CoA ligase (4CL)), followed by *m*-hydroxylation by C3H to yield caffeoyl-CoA. The hydrolysis of caffeoyl-CoA by CoA thioesters results in the production of caffeic acid and CoA [19]. The enzymes involved in this biosynthetic pathway have been elucidated through knockout mutations and RNAi-mediated studies in *Arabidopsis* plants, facilitating the identification and functional characterization of key genes and enzymes [20,21]. Figure 3 shows a schematic representation of the biosynthetic pathway.

## 4. Natural Sources with High Content of CA

CA is abundantly present in a wide range of plant-derived foods, including fruits, vegetables, and certain beverages. The concentration of CA within a particular plant species varies depending on factors such as the plant part (e.g., fruits, leaves, stems, and roots), degree of ripeness, processing techniques, and storage conditions. Natural sources rich in CA include coffee beans, especially when freshly roasted, which can contain significant amounts of CA depending on the coffee type and brewing method [22,23]. Fruits such as apples, pears, cherries, and grapes are known to contain CA [24,25], while vegetables like bell peppers, broccoli, carrots, and spinach also contribute to CA intake [26,27]. Herbs and spices like thyme, oregano, sage, rosemary, and basil are recognized as rich sources of CA [28,29]. Additionally, whole grains like wheat, oats, and rice, particularly their bran and outer layers, contain this polyphenol [30,31]. Recent research has revealed the presence of CA in certain mushrooms, including *Agaricus bisporus*, *Coprinus atramentarius*, *Morchella elata*, and *Laetiporus sulphureus* [32]. Furthermore, the traditional consumption of artichoke (*Cynara scolymus*) has been identified as a significant source of CA, particularly in its leaves [33,34]. The Phenol-Explorer database (accessed on 12 March 2024) (http://phenol-explorer.eu/) offers comprehensive information on polyphenol levels in various foods, comprising over 35,000 data points encompassing 500 distinct polyphenols across more than 400 food items. Table 1 provides an overview of the major natural sources of CA. 

## 5. Absorption, Distribution, and Metabolism of CA

The potential therapeutic use of CA relies heavily on its pharmacokinetic behavior, which encompasses its stability within the digestive system, including exposure to acidic pH, bile, and metabolic enzymes, as well as its absorption in the intestines, distribution throughout the body, and subsequent metabolic processes leading to excretion. Studies have demonstrated that CA can traverse the blood–brain barrier (BBB) and reach concentrations of 0.02 µM in the cerebrospinal fluid, thereby exerting beneficial effects on the brain [44]. To investigate the pharmacokinetics of CA, experiments were conducted in rats using radioactive-labeled CA [3-^14^C]. Rats were orally administered 1.52 mg of labeled CA [3-^14^C] (140 × 10^6^ dpm) via gavage, and various samples, including organs, tissues, plasma, urine, and feces, were collected over a duration of up to 72 h. These samples were then analyzed using high-performance liquid chromatography (HPLC) coupled with online radioactivity detection and tandem mass spectrometry, allowing for the determination of residual radioactivity in the samples. The results of these analyses are summarized in Table 2.

One hour after ingestion, approximately 80% of the radioactivity remained in the gastrointestinal (GI) tract, while the remaining 20% had already entered the bloodstream and kidneys. This indicates that the absorption of CA starts in the stomach and is subsequently excreted, primarily through urine, in the form of nine identified radioactive compounds, including *trans*-caffeic acid, *cis*-caffeic acid, caffeic acid 3′-*O*-sulfate, caffeic acid-4′-*O*-sulfate, caffeic acid 3′-*O*-glucuronide, caffeic acid 4′-*O*-glucuronide, isoferulic acid-3′-*O*-sulfate, ferulic acid-4′-*O*-sulfate, and ferulic acid-4′-*O*-glucuronide (refer to structures in Figure 4), along with four unidentified metabolites. By the 72-h mark, there was minimal to no accumulation of radioactivity in various body tissues, including the kidney, brain, testes, lung, heart, muscle, liver, spleen, and red blood cells. Only a small quantity of an unidentified ^14^C-labeled metabolite was excreted in feces. The overall recovery of radioactivity was approximately 80% [45].

Kishida and Matsumoto conducted a study on the metabolism of CA in male Wistar rats that were administered a dosage of 40 mg/kg. They found that at 48 h after administration, approximately 61.6% of CA was excreted, with the excretion primarily consisting of glucuronidated and/or sulfated CA-conjugates. To analyze these conjugates and quantify the amount of free CA, specific enzymes were utilized to release CA from the conjugates. Subsequently, CA levels in the urine were measured using HPLC-DAD [46].

## 6. Signaling Pathways Affected in Cancer Progression

Despite continuing advances in treatments, cancer remains a major health challenge, with cardiovascular diseases and cancers being the most common causes of death worldwide [47]. Understanding the detailed mechanisms of carcinogenesis, which are triggered by prolonged exposure to various risk factors, is crucial for preventing cancer development and progression [48]. Hanahan and Weinberg have summarized several cellular processes that contribute to the emergence of neoplasms and their malignant progression, including self-sufficiency in growth signals, insensitivity to growth-inhibitory signals, evasion of apoptosis, unlimited replicative potential, tissue invasion and metastasis, and sustained angiogenesis [49].

Although cancer is a multifactorial disease, a common outcome of exposure to risk factors is inflammation and uncontrolled production of ROS. ROS and RNS are a group of radical and non-radical molecules produced by cellular metabolism or induced by external sources. Excessive ROS formation can damage macromolecules such as DNA, proteins, and lipids, leading to genomic instability and altered cell growth [50]. ROS can influence cell cycle progression by affecting the activity of proteins like cyclin-dependent kinase inhibitor p21 or the serine/threonine mutant ataxia telangiectasia protein kinase (ATM). ATM is crucial for DNA repair and impacts cell signaling pathways that are important for proliferation and apoptosis, such as the Akt or p53 pathway [48]. Normally, intracellular ROS accumulation due to carcinogen exposure is mitigated by antioxidant enzymes like catalase (CAT), superoxide dismutase (SOD), or the glutathione (GSH) system. However, when ROS levels exceed the cell’s antioxidant capacity, extensive oxidative damage to cellular components occurs, increasing the likelihood of mutations in oncogenes or tumor suppressor genes and leading to the formation of precancerous lesions. Consequently, ROS are attributed to a tumor-promoting role during carcinogenesis as they can inactivate or activate proteins involved in cancer-related signaling pathways.

ROS, chronic infections, and inflammation all activate the NF-κB pathway, promoting the dimerization of the inhibitor of κB kinase (IKK) gamma (IKKγ), also known as the nuclear factor κB (NF-κB) essential modulator (NEMO). This component of the IKK complex is crucial for the canonical activation of the NF-κB pathway [51,52,53]. The dimerization of IKKγ/NEMO triggers the phosphorylation of inhibitory NF-κB proteins (IκBs), leading to their degradation by the proteasome. This process induces the phosphorylation of p50/p65 dimers via the activation of protein kinase-A (PKAc), facilitating the translocation of NF-κB to the nucleus, where it regulates the transcription of genes associated with survival, cell proliferation, proinflammatory cytokines, and ROS-related genes [54,55]. The activation of the NF-κB signaling pathway is linked to various cellular processes, including inflammation and immune response. The disruption of this pathway has been associated with inflammatory diseases and the development and progression of cancer. In cancer, specifically, NF-κB activation is linked to apoptosis resistance as it negatively regulates anti-apoptotic proteins B-cell leukemia/lymphoma 2 (Bcl-2) and B-cell lymphoma-extra large (Bcl-XL). It is also linked to increased malignancy by modulating the transcription of genes related to cell proliferation of cyclin D1 (CCND1) and cellular Myc (c-Myc). It also influences the expression of genes involved in angiogenesis, such as vascular endothelial growth factor (VEGF), proinflammatory cytokines related to carcinogenesis, such as IL-1β and IL-6, genes involved in detoxification and drug resistance, and antioxidant enzymes such as SOD, CAT, thioredoxin (TRX), HO-1, and glutathione peroxidase (GPX1) [54].

Another pathway activated by ROS is the Nrf2 pathway. Nrf2 is a key regulator of cellular redox homeostasis, controlling the expression of various antioxidant and cytoprotective genes involved in drug detoxification, cell proliferation, metabolism, autophagy, apoptosis, proteasome function, DNA repair, and antioxidant response, collectively known as phase II genes. Under normal conditions, Nrf2 associates with the Kelch-like Echassociated protein 1 (Keap1), part of the Cullin 3 (CUL3)-based ubiquitin ligase complex that regulates Nrf2 stability, and promotes its degradation via the ubiquitin/proteasome pathway. However, in the presence of electrophilic molecules or oxidative stress, cysteine residues in Keap1 are modified, leading to its inactivation and reducing its binding to Nrf2 or CUL3, thereby preventing Nrf2 degradation. Stabilized Nrf2 translocates to the nucleus, where it dimerizes with other leucine zipper proteins of the v-maf avian musculoaponeurotic fibrosarcoma oncogene homolog (Maf) family to activate AREs on phase II genes. Additionally, Nrf2 degradation can be mediated by β-transducin repeat-containing protein (β-TrCP) in the nucleus, which binds to phosphorylated Nrf2 and Cullin 1 (CUL1), inducing Nrf2 ubiquitination. Another mechanism involves Keap1 degradation mediated by phosphorylated p62/Sequestosome-1 (p62/SQSTM1) [54,56]. Strong cellular antioxidant, drug-detoxifying, and cytoprotective activities have been linked to cancer malignancy, with evidence showing that cancer cells often exhibit aberrant Nrf2 activation. Various mechanisms have been identified, including somatic mutations in Nrf2, Keap1, or CUL3 genes, epigenetic silencing of Keap1, accumulation of Keap1-interacting proteins like p62/SQSTM1 or p21, and cysteine modification by oncometabolites such as fumarate, which affect Keap1 activity [54]. Nrf2 activation has been associated with resistance to chemotherapy and radiotherapy and poor prognosis in cancers such as head and neck, lung, ovarian, and breast cancer [57,58,59,60,61,62,63]. Importantly, several natural and chemical compounds are being tested to inhibit Nrf2 signaling in cancer cells with high Nrf2 activity, while Nrf2 inducers are being explored to protect normal cells from carcinogens [64].

Oxygen deprivation in tumors significantly influences cancer development and metastasis. Hypoxia-inducible transcription factors (HIFs) are crucial in promoting the survival, proliferation, adaptability, and motility of malignant cells. Additionally, hypoxia elevates Programmed death-ligand 1 (PD-L1) expression, also known as cluster of differentiation 274 (CD274) or B7 homolog 1 (B7-H1), in both malignant and immunoregulatory cells [65]. ROS also play a role by oxidizing prolyl hydroxylase domain (PHDs) proteins, essential for the binding of von Hippel–Lindau tumor suppressor protein (p-VHL) to HIF-1α, leading to its ubiquitination and degradation in the proteasome. HIF-1 operates as a heterodimeric complex composed of an alpha and a beta subunit. HIF-1β is constitutively expressed, while HIF-1α accumulation is induced under hypoxic conditions. In normoxia, PHDs hydroxylate two prolyl residues of HIF-1α, marking it for ubiquitination and subsequent von Hippel–Lindau (VHL) complex-mediated degradation [54]. Conversely, during hypoxia, HIF-1α accumulates, and the HIF-1 complex activates the transcription of genes containing hypoxia response elements (HREs) in the cell nucleus. HIF-1 signaling has been extensively studied in various cancers, including pancreatic, gastric, and prostate cancers, due to its regulation of genes involved in angiogenesis, metabolism, glucose transport, and cell migration [66,67,68,69,70].

In neoplastic cells, the ErbB family of proteins and their signaling pathways are frequently altered. This family consists of four receptor tyrosine kinases related to the epidermal growth factor receptor (EGFR); the name of the family is derived from the viral oncogene homologous to erythroblastic leukemia viral oncogene. In humans, this family includes Human epidermal growth factor receptor 1 or Her 1 (EGFR, ErbB1), Her2 (ErbB2), Her3 (ErbB3), and Her4 (ErbB4) [71]. ErbB signaling influences cell proliferation, migration, differentiation, apoptosis, and motility via the Phosphoinositide 3-kinase/serine threonine-protein kinases (PI3K/Akt), the Janus kinase/signal transducer and activator of transcription (JAK/STAT), and Mitogen-activated protein kinase (MAPK) pathways. Aberrant signaling of ErbB family members is critical in tumorigenesis and immune evasion in various malignancies. Excessive ErbB signaling is linked to the development of numerous solid tumors, and direct alterations in these receptor-dependent pathways are associated with cancer progression [72].

The PI3K/AKT pathway is pivotal for cell survival, proliferation, and protein biosynthesis. Phosphoinositide 3-kinase (PI3K) is a family of lipid kinase enzymes that phosphorylate the 3’-OH group of phosphatidylinositols (PtdIns) in the plasma membrane (Class I) or intracellular membranes for vesicular trafficking regulation (Classes II and III). These proteins are biologically significant as their activation influences cellular processes such as growth, survival, metabolism, inflammation, motility, proliferation, and therapy resistance [73]. Class I PI3K and its downstream effector AKT/PKB are activated by extracellular growth factors like epidermal growth factor (EGF), platelet-derived growth factor (PDGF), or insulin, triggering a phosphorylation cascade that culminates in the activation of the mammalian target of rapamycin (mTOR) kinase and the Ras/MAPK pathway. Ras proteins, from the acronym Rat sarcoma virus, are located in the plasma membrane of cells and act as molecular switches that send signals to activate cell growth. The Ras family is a protein superfamily of small GTPases. Members of the superfamily are divided into families and subfamilies based on their structure, sequence, and function. The five main families are Ras, Ras homologous (Rho), Ras-related nuclear protein (Ran), Ras-related in brain (Rab), and ADP-ribosylation factor (Arf) GTPases. The activation of the PI3K/AKT pathway is widely associated with carcinogenesis and is frequently observed in various human cancers [74,75,76,77]. High levels of ROS can oxidize and inactivate the phosphatase and tensin homolog (PTEN), promoting PI3K/AKT pathway activation [54]. PTEN acts as a phosphatase to dephosphorylate PtdIns, counteracting PI3Ks by specifically catalyzing the dephosphorylation of the 3phosphate of the inositol ring at phosphatidylinositol 3,4,5-trisphosphate (PIP3) to produce phosphatidylinositol 3,4-bisphosphate (PIP2). This dephosphorylation is crucial because it inhibits the Akt signaling pathway; thus, sustained PTEN inactivation leads to excessive PI3K/AKT pathway activation [78]. The PTEN phosphatase is encoded by the PTEN tumor suppressor gene, and mutations or deletions of this gene contribute to constitutive PI3K/AKT signaling, facilitating the development of cancers such as glioblastoma, lung cancer, breast cancer, and prostate cancer [79,80,81,82,83]. Clinical trials are currently investigating PI3K/AKT pathway inhibition for treating pancreatic, lung, ovarian, breast, melanoma, and hematological malignancies [54].

Mitogen-activated protein kinases (MAPKs) are a group of enzymes that phosphorylate various cellular proteins, including transcription factors, nuclear proteins, membrane transporters, cytoskeletal elements, and other kinases, upon activation. This phosphorylation regulates crucial cellular processes, such as proliferation, migration, differentiation, senescence, and cell death. MAPKs are activated by extracellular signals like growth factors (EGF, PDGF, insulin), cytokines, and intracellular stressors. Key MAPK subfamilies, such as ERK1/2, Jun N-terminal kinase (JNK), and p38, respond to different stimuli. The Ras/Raf/MEK/ERK1/2 pathway is notably relevant in cancer, often altered by excessive growth factors, hormonal signaling, or oncogenic mutations, leading to abnormal mitogenic signaling, increased proliferation, and apoptosis resistance. Raf is an acronym for Rapidly Accelerated Fibrosarcoma and is the best characterized Ras effector and a member of a family of serine/threonine kinases. Mutations in Ras family genes, found in about 30% of human cancers, activate proteins like PI3K, Rac, and Rho, which influence cytoskeletal dynamics and cell invasiveness. Rac-GTPase represents a subfamily of the Rho family. Specific small molecule inhibitors targeting the Ras/MAPK pathway are under clinical trials, including those for the KRAS-G12C mutation, with promising clinical outcomes anticipated. The Ras/MAPK pathway activation is intricately regulated and closely linked to ROS production. ROS can directly activate MAPKs without external stimuli like EGF, and antioxidant inhibition has been shown to prevent this activation in various experimental setups. Ras-mediated transformation involves NADPH oxidase 1 (NOX1) activation via the Ras-Rac1-NOX1 pathway. Key redox sensors in the MAPK cascade include TRX and apoptosis signal-regulating kinase 1 (ASK1). TRX inhibits ASK1 when reduced, but ROS-induced oxidation releases ASK1, activating JNK and p38 signaling, which affects cell proliferation, differentiation, apoptosis, inflammation, and stress response. Additionally, ROS can activate ERK signaling, potentially through the upstream activation of the EGF receptor (EGFR). This process involves NOX-mediated H2O2 generation and SHP-2 phosphatase inactivation, sustaining EGFR activation and promoting cell signaling via the Ras-Raf-MEK-ERK pathway, impacting cell proliferation and differentiation. MAPK activation is also linked to mitochondrial changes, including increased mitochondrial ROS production. Mutations in Ras or ERK2 activation can induce mitochondrial fission and increase mitochondrial mass, affecting ATP production and contributing to cancerous traits in cells with Ras mutations [54].

The intracellular JAK/STAT pathway and modification of histone marks on nucleosomes regulate the expression of proinflammatory mediators, playing a crucial role in carcinogenesis [84]. The activation of the JAK/STAT pathway occurs in response to various hormones (prolactin, growth hormone, leptin, and erythropoietin), cytokines, and growth factors through their respective receptors. This signaling pathway regulates cell proliferation, differentiation, survival, motility, and apoptosis in different tissues [85]. STATs are latent cytoplasmic transcription factors activated by phosphorylation through Janus kinase (JAK). The JAK family includes JAK1-3 and tyrosine kinase 2 (TYK2), which interact non-covalently with membrane receptor domains or intracellular portions of growth factor receptors. Upon cytokine binding, JAKs phosphorylate tyrosine residues to transmit signals from membrane-bound receptors. Phosphorylated STAT molecules then dimerize and translocate to the nucleus, initiating gene transcription that regulates cell cycle progression, apoptosis initiation, and occurrence. These processes are vital for cellular homeostasis, and disruptions in this pathway contribute to cancer progression, inflammatory diseases, and autoimmune disorders. STAT family proteins are implicated in creating and sustaining a procarcinogenic inflammatory microenvironment, initiating malignant transformation, and promoting cancer progression. Inflammation plays a critical role in tumor initiation and progression, both as an initiator of oncogenic transformations and as a promoter through genetic and epigenetic alterations that foster an inflammatory microenvironment conducive to tumor growth. Additionally, immune responses and inflammatory mediators mediated by STAT3, STAT5, and STAT6 suppress antitumor immunity, further contributing to carcinogenesis. Given the JAK-STAT pathway’s role in regulating cell cycle and apoptosis effector molecules, it is evident that this pathway promotes carcinogenesis by enhancing proliferation and modulating programmed cell death [84]. 

In human cancer, the TP53 tumor suppressor gene, situated on chromosome 17, is the most frequently mutated. It encodes p53, a transcription factor binding to specific DNA sequences within the genome, activating adjacent genes and those controlled by enhancers with p53 binding sites. Additionally, p53 can repress the transcription of numerous genes, typically through indirect mechanisms [86]. The primary biological role of p53 is well-understood: it induces cell cycle arrest, senescence, or apoptosis in response to DNA damage, thereby preventing the accumulation of oncogenic mutations, earning it the nickname “guardian of the genome”. Moreover, p53 participates in other cellular processes like autophagy, metabolism, and cellular plasticity [87]. It enhances glutamine catabolism, supports antioxidant activity, reduces lipid synthesis, promotes fatty acid oxidation, and stimulates gluconeogenesis. Under normal conditions in unstressed cells, p53 levels remain low due to continuous proteasomal degradation mediated by the Mouse double minute 2 homolog (MDM2), an E3 ubiquitin ligase, and the principal inhibitor of p53, along with COP1 and p53-induced RING-H2 protein (Pirh2). Additionally, transformed mouse 3T3 cell double minute 4 (MDM4), formally named MDMX, restricts p53’s biochemical activity as a transcription factor, serving as a physiological inhibitor [86]. At basal levels, p53 maintains the transcription of several antioxidant genes (SESN1/2, GPX1, and AIF). However, during stress conditions, p53 accumulates in the cytoplasm and induces the expression of NQO1 and proline oxidase (POX), enzymes that generate ROS, alongside Bcl2-associated X, apoptosis regulator (Bax), and p53-upregulated modulator of apoptosis (PUMA), which promote mitochondrial uncoupling and ROS production, culminating in oxidative stress and apoptosis. Bax and Bcl-2 homologous antagonist/killer (Bak) are members of the Bcl-2 family and core regulators of the intrinsic pathway of apoptosis. Upon apoptotic stimuli, they are activated and oligomerized at the mitochondrial outer membrane (MOM) to mediate its permeabilization, which is considered a key step in apoptosis. PUMA is a proapoptotic homolog domain-3-only member of the Bcl-2 protein family, which has been demonstrated to be critical for some cells’ apoptosis induced by ER stress. A crucial target of p53 involved in ROS metabolism and regulation is the TP-53-induced regulator of apoptosis and glycolysis (TIGAR), which inhibits glycolysis and enhances glucose flux into the pentose phosphate pathway, a major source of NADPH utilized in reducing GSH. Conversely, p53 mutations in cancer cells impair mitochondrial respiration and elevate glycolysis [54].

In summary, multiple pathways and factors contribute to the emergence and progression of various cancers. Investigating the dysregulation of intracellular pathways in cancer and exploring molecules capable of restoring these pathways are crucial endeavors.

## 7. Anti-Inflammatory Activity of Caffeic Acid against Cancer

As mentioned in the previous section, chronic inflammation has been linked to the initiation and progression of cancer [88,89,90]. ROS and RNS, generated during inflammation, can inflict DNA damage, potentially leading to mutations that contribute to the maintenance of tumors [91,92]. Chronic inflammation often triggers cellular proliferation as part of tissue repair mechanisms, inadvertently creating conditions favorable for the expansion of cells with damaged DNA or mutations, thus heightening cancer risk. Moreover, inflammation has been implicated in cancer progression by promoting the production of inflammatory cytokines that support cell proliferation, inhibit apoptosis, stimulate tumor angiogenesis and vascularization, and facilitate tumor invasion and metastasis.

NF-κB is a crucial protein complex responsible for controlling DNA transcription, cytokine synthesis, and cell viability. The dysregulation of NF-κB is associated with immune response irregularities and cancer advancement, making it a significant factor in tumor biology. It influences pathways like JNK/p38 MAPK and positively regulates c-Myc, thus playing a role in various reported neoplasms. Consequently, the inhibition of NF-κB holds promise as a target for cancer therapy development [93]. In this context, CA has shown the capability to regulate this pathway, thereby disrupting cancer cell proliferation, viability, and invasion in endothelial cells, triggered by lipopolysaccharide (LPS), as well as in mammary epithelial cells [94,95]. Consequently, CA modulation of cytokines may affect the production and function of various inflammatory cytokines, including tumor necrosis factor-alpha (TNF-α) [96] and interleukins IL-1β, IL-6, and IL-8 [97,98,99,100].

CA has been discovered to impede Cyclooxygenase-2 (COX-2), an enzyme pivotal in the inflammatory cascade. Elevated COX-2 expression is evident during prolonged inflammation, where it can foster the onset and progression of chronic ailments, such as cancer. Studies suggest that CA, at concentrations ranging from 50 to 10 µM, effectively suppresses COX-2 and its byproduct, prostaglandin E2 (PGE2), in human colon myofibroblast cells (CCD-18Co) [99]. Consequently, CA also hampers the enzyme Nitric Oxide Synthase (iNOS), curtailing nitric oxide (NO) production, which may aid in inhibiting tumor angiogenesis [101].

Inflammation induces the secretion of molecules like VEGF, which stimulates the development of new blood vessels that are crucial for supplying oxygen and nutrients to tumors. The activity of VEGF is amplified by HIF-1α and STAT3. In a Caki-I carcinoma animal model, CA was observed to suppress STAT3 phosphorylation, thus diminishing its function and consequently impeding HIF-1α, resulting in reduced tumor vascularization and VEGF gene expression [16]. In retinal endothelial cells, CA demonstrated potent anti-angiogenic effects, mitigating vaso-proliferative retinopathies associated with ROS-induced VEGF [102]. In hepatocellular carcinoma cells (HCC), CA curbed VEGF and vascularization induced by CoCl_2_, accomplishing this through the stabilization of STAT3/JNK1/HIF-1α [103]. Guo et al. discovered that CA serves as a potent inhibitor of prolyl hydroxylase-2 (PHD2), an enzyme responsible for HIF hydroxylation and degradation. In neuronal cells such as PC12 and SH-SY5Y, CA attenuated cell apoptosis induced by hypoxia. Furthermore, in mice, CA administration decreased injury markers such as lactate dehydrogenase, malondialdehyde, and lactic acid [104]. 

Another process associated with inflammation in cancer is the production of enzymes such as metalloproteinases (MMPs), which play a role in breaking down the extracellular matrix of cells. MMP-2 and MMP-9 can degrade these barriers, facilitating the invasion of cancer cells into nearby tissues and their spread to distant areas. Through computational studies, CA has been identified as an inhibitor of MMP-9 and dipeptidyl peptidase-4 (DPP-4) enzymes, with respective IC_50_ values of 158.19 and 88.99 µM [105]. Chung et al. explored the enzymatic inhibitory effects of CA extracted from *Euonymus alatus* and its synthetic derivative caffeic acid phenethyl ester (CAPE) in animal experiments using rats. The animals were administered CA and CAPE at doses of 5 mg/kg subcutaneously or 20 mg/kg orally. Both compounds exhibited the inhibition of MMP-2 and -9, with no effect on MMP-1, -3, and -7. Moreover, in xenograft models, CA and CAPE hindered the growth of HepG2 tumors and liver metastasis by suppressing MMP-9 activity and decreasing NF-κB expression [106]. These multifaceted mechanisms orchestrated by CA in cancer progression are depicted in Figure 5.

## 8. Caffeic Acid Properties against Cancer

Apoptosis, a regulated cell death mechanism, serves to eliminate damaged or unnecessary cells. CA has demonstrated the ability to trigger apoptosis in cancer cells by influencing the mitochondrial pathway. This is accomplished by the inhibition of the Bcl-2 family of proteins, leading to an increase in pro-apoptotic members (such as Bax and Bak) and a decrease in anti-apoptotic ones (such as Bcl-2 and Bcl-xL) [107]. This action can result in the release of cytochrome c from the mitochondria into the cytosol, initiating the activation of caspases—protease enzymes crucial for programmed cell death [108]. Specifically, CA at a concentration of 10 µg/mL can activate both initiator caspases (like caspase-9) and executioner caspases (such as caspase-3 and caspase-7), leading to the breakdown of cellular components necessary for apoptosis (Figure 4) [109]. Furthermore, CA has been found to induce the expression of p53, a tumor suppressor protein pivotal in preventing cancer development [110]. The activation of p53 can result in cell cycle arrest and apoptosis.

In general, it has been shown that CA is a potent modulator of apoptosis and autophagy in cancer cells, thus affecting their proliferation and survival, including in carcinoma cells in the head and neck [111] and MCF-7 [112], multiple myeloma (MM.1R, RPMI8226, and U266) [113], leukemia (K562) [114], and osteosarcoma cells (MG-63) [115]. In human melanoma SK-Mel-28 cells, the administration of CA (0–200 μM) induces apoptosis and cell cycle arrest by increasing the expression profile of caspase 1 and caspase 3 [116]. Moreover, CA (200–800 μM) has been shown to promote Ca^2+^ accumulation in cells in a concentration-dependent manner, an effect that is closely related to apoptosis. CA releases Ca^2+^ from the endoplasmic reticulum by inducing protein phospholipase C and then induces apoptosis in SCM1 gastric cancer cells [117]. In vivo, the effect of novel caffeic acids conjugated with silver nanoparticles with and without gamma radiation exposure has been found to be effective against experimentally induced Ehrlich tumors, resulting in growth inhibition in solid tumor cells, whose underlying mechanism involves an apoptotic effect [118].

The impact of CA on mitochondria has been explored, particularly in breast cancer cells like MCF-7 and MDA-MB-468 cell lines. Treatment with CA at a concentration of 20 μM disrupts mitochondrial function, which leads to several effects: increased Caspase-9 activity, elevated levels of ROS, and a decrease in membrane potential (Δψm) [119]. Additionally, it has been suggested that CA may specifically influence protein kinase C delta (PKCδ), a protein that is involved in apoptosis and affects mitochondria by reducing ΔΨm. Consequently, CA promotes apoptosis in MG-63 osteosarcoma cells by inducing the translocation of PKCδ to mitochondria and reducing ΔΨm, resulting in mitochondrial membrane potential (MMP) alterations [115]. Additionally, an antioxidant derived from CA, known as AntiOxCIN6, has been developed. This compound is formed by linking the antioxidant core to the lipophilic TPP+ via a 10-carbon aliphatic chain. Interestingly, AntiOxCIN6 has demonstrated the potential to indirectly induce apoptosis. Despite increasing the antioxidant defense system, this complex is inefficient in eliminating ROS. Consequently, it affects mitochondrial function by impairing its ATP production capacity. Although this impairment does not directly affect cell viability, it sensitizes A549 adenocarcinoma cells to apoptotic death induced by cisplatin [120]. A summary of both the anti-inflammatory and proapoptotic anticancer activity of CA can be seen in the flowchart in Figure 6. Here, it is observed that CA affects signaling pathways that result in cell death, decreased proliferation, and migration of cancer cells, as well as a decrease in angiogenesis processes.

Regarding the modulation of CA in oncogenic signaling pathways in cancer, it has been reported that the administration of CA (100 μM) alone or in combination with metformin (10 mM) is efficient in stimulating the AMPK signaling pathway, which acts by preventing de novo synthesis of unsaturated fatty acids, consequently reducing cancer cell survival [121]. AMP-activated protein kinase (AMPK) is a vital enzyme in regulating cellular metabolism and maintaining energy balance. It also plays a significant role in tumor progression by influencing the expression of genes associated with invasion, metastasis, and angiogenesis. Consequently, activators of AMPK hold promise for exerting antitumor effects by augmenting sensitivity to chemotherapy and radiotherapy while mitigating metastasis [122]. CA has also been found to disrupt the PI3K/Akt signaling pathway, both in laboratory studies and in living organisms. This interference shows promise in inhibiting the proliferation and self-renewal ability of cancer stem cells (CSCs). Such effects have been investigated in CSC populations derived from the human colon adenocarcinoma cell line HCT116 [121,123]. In melanoma, TGFβ is one of the key signaling pathways in progression because, in some circumstances, it can provide the ideal microenvironment for tumor development. In this sense, a study showed that CA (1 mmol/L), together with a moderately potent SMF (0.7 T), decreased the expression of TGFβ, which could support anticancer therapy [124]. Conversely, tissue transglutaminase type 2 (TG2) is an enzyme involved in apoptosis, wherein its activity, among other roles, stimulates caspase activation, a process often induced by oxidative stress. In this context, a study conducted by Feriotto et al. revealed that CA effectively stimulates the activation pathways of TG2. This was shown by the antiproliferative impact observed in the K562 cell line (chronic myeloid leukemia), which correlated with TG2 activity. Such activation of TG2 contributed to the elevation of ROS levels, consequently leading to cell death [114].

In hepatocellular carcinoma (HCC), CA inhibits the activity of GRP75 (75 KDa Glucose-Regulated Protein) induced by low doses of the carcinogen Benzo(a)pyrene (B[a]P) through both transcriptional and post-transcriptional modifications. This inhibition is closely linked to CA’s role as an inhibitor of NF-κB, which serves as an upstream transcriptional regulator of GRP75. Additionally, CA induces the cleavage of the GRP75–p53 complex, leading to the nuclear translocation and activation of p53. This promotes the induction or maintenance of anti-apoptotic capacity and multidrug resistance (MDR) characteristics in HCC. Given its pivotal role, GRP75 is implicated in the onset and progression of cancer [125].

## 9. Caffeic Acid as an Adjuvant for Chemotherapy

Drug resistance represents a significant obstacle to the success of chemotherapy in treating many types of human tumors, and the extent of its effect varies depending on factors such as the cancer type, the specific drug used, and patient-related factors [126]. One mechanism through which cancer cells develop resistance to chemotherapy involves the overexpression of drug efflux pumps, such as P-glycoprotein (P-gp), which diminishes the intracellular concentration of chemotherapeutic drugs. CA has demonstrated the ability to regulate the activity of these pumps, potentially mitigating drug efflux and enhancing the retention of chemotherapeutic agents within cells, thus improving the effectiveness of cancer treatments. Research conducted by Teng et al. investigated the impact of CA on P-gp both in vitro and in silico. Their findings revealed that the combination of CA with chemotherapeutic agents like doxorubicin reduced the activity of P-gp, as indicated by lower IC_50_ values in resistant cancer cells. Additionally, CA inhibited the efflux of rhodamine 123. In silico analysis suggested that CA binds to P-gp through polar interactions with specific residues, including Glu74 and Tyr117 [127]. A search and compilation of studies where CA was tested as a coadjuvant for the treatment of various types of cancer are presented below, divided into in vitro trials, in vivo trials, and clinical trials.

### 9.1. In Vitro and In Vivo Trials

CA exhibits adjuvant effects, enhancing the apoptotic response when combined with cisplatin in various cancer cell lines such as Lacks’ cervical cancer cells (HeLa), cervical cancer (CaSki), non-small cell lung cancer cell (A549), and hepatoblastoma cell line (HEPG2) [128,129]. Additionally, Sirota et al. demonstrated that the combined treatment with cisplatin (5 µM) and CA (10 µM) restored the chemo-sensitizing effect against cisplatin-resistant ovarian endometrioid adenocarcinoma cells (A2780). This combined treatment resulted in a reduction of cell viability comparable to that achieved in sensitive cells treated solely with cisplatin at the same concentration. However, it was noted that pre-incubation with CA before cisplatin treatment could induce resistance by activating Nrf2. This highlights the importance of cautious examination when using CA as an adjuvant to cisplatin [130]. On another note, the cocrystallization of CA with 5-fluorouracil (5-FU) has been found to enhance the physicochemical properties of 5-FU, such as solubility and tissue permeability. This synergistic interaction leads to an improved anticancer activity of 5-FU, as indicated by a combination index (CI) of less than 1 [131]. Table 3 summarizes the results of in vitro and in vivo co-treatments using CA against cancer cells.

In a recent study examining the impact of caffeic acid on lung cancer cells, it was observed that CA effectively inhibits proliferation, migration, and apoptosis by targeting the TMEM16A protein, which is a calcium-activated chloride channel. The inhibitory concentration (IC_50_) of CA for TMEM16A was determined to be 29.47 ± 3.19 μM. Further animal studies corroborated these findings, demonstrating a significant reduction in tumor size. Notably, when combined with 5.4 mg/kg caffeic acid and hydroxydaunorubicin (DOX) at a dosage of 4.1 mg/kg, the treatment resulted in an 85.6% reduction in tumor size and minimized adverse effects. In conclusion, the combined therapy of caffeic acid and DOX proved more effective in inhibiting lung cancer cell growth compared to either drug administered alone, even at higher doses [132].

In addition to the re-sensitization of cancer cells, a protective effect of CA towards other organs has also been seen during the use of chemotherapeutics. For example, the effect of CA encapsulated in a nanoemulsion on the reduction of nephrotoxicity towards non-cancerous cells of the HEK 293 renal line was evaluated, showing an improvement in cell viability in renal cells from 33% to more than 95% during treatment with cisplatin [129]. This finding and similar results are also summarized in Table 3.
ijms-25-07631-t003_Table 3Table 3Synergetic effect of CA in cancer cells.AimCancer Type: ModelTreatment ConditionsFindingReference
To assess the efficacy of cisplatin + CA treatment in human cervical cancer

• In vitro:human cervical cancer cell lines: HeLa, SiHa, CaSki (HPV-positive), and C33A (HPV-negative) cells.

CA (300 µM) and cisplatin (11 µM) for 24 h

The combination of cisplatin and CA significantly inhibited cell growth of HeLa and CaSki cell lines, with a combination index < 1, indicating a synergistic effect. The combination significantly increased the expression of caspases 3, 7, and 9, demonstrating apoptosis.
[128]
To assess the efficacy of the combined treatment with cisplatin + CA against ovarian carcinoma

• In vitro: ovarian carcinoma cells A2780 and ovarian carcinoma-resistant A2780cisR cells.

CA (10 µM) and cisplatin (5 µM) for 24 h

The combined therapy restores the sensitivity of resistant cells to cisplatin, achieving a similar level of cell viability as that observed in sensitive cells (60% viability). When the cisplatin/caffeic acid ratio was increased to 1:10 (5:50 µM), the caspase activity rose significantly by 4.3-fold.
[130]
To evaluate the effects of metformin (Met) and CA on metastatic human cervical cancer

• In vitro:metastatic human cervical HTB-34 cell line.

CA (100 µM) and Met (10 mM) for 24 h

CA (100 µM) and Met (10 mM) activated AMPK. CA increased oxidative stress, affecting bioenergetics pathways and making HTB-34 cells more sensitive to Met. CA and Met suppressed HTB-34 cells by different mechanisms.
[133]
To determine the efficacy and underlying mechanisms of CA in combination with paclitaxel for the treatment in human non-small cell lung carcinoma (NSCLC)

• In vitro:human non-small cell lung carcinoma H1299 cells. • In vivo:mouse xenograft model by subcutaneous injections of H1299 cells.

In vitro: 100 μM CA + 10 μM of paclitaxel for 24 hIn vivo: 20 mg/kg CA and 10 mg/kg paclitaxelad ministered concomitantly for three weeks.

In vitro, combination treatment decreased the proliferation of NSCLC H1299 cells by the MAPK pathway. CA induced sub-G1 cell cycle arrest in H1299 cells.In vivo, the combined treatment with CA and paclitaxel exerted a more effective suppressive effect on tumor growth in H1299 xenografts without causing significant adverse effects.
[134]
To evaluate the synergistic antitumor activity and the physicochemical and pharmacokinetic properties of caffeic acid/5-FU-cocrystal in vitro and in vivo.

• In vitro: human colon cancer HCT-116, breast cancer MDA-MB-231, and lung cancer A549 cell lines• In vivo: Sprague Dawley rats

In vitro: HCT-116; MDA-MB-231 (15.19 μM); and A549 (11.57 μM) of caffeic acid/5-FU cocrystal for 48 h.In vivo: oral dose of 50 mg kg^−1^.

In vitro: Cocrystallization of CA + 5-FU optimized the physicochemical properties of 5-FU and exerted a synergistic antitumor effect (CI < 1), thus enhancing the anticancer activity of 5-FU.In vivo: The aqueous solubility and permeability of 5-FU in the cocrystal increased by 1.92 and 4.22-fold, respectively, compared to the original drug 5-FU.
[131]
To evaluate the effects of CA and imatinib (IM) and their synergistic effects on chronic myeloid leukemia model

• In vitro:human myelogenous leukemia cell line K562 and (IM)-resistant cells.

Synergistic effects of CA (up to 38 µM) and IM (up to 0.15 µM) on K562 cells.
CA induced apoptosis in IM-resistant cells and reduced their proliferation. Combination treatment with CA and IM showed synergistic effects, increasing the antiproliferative activity.[114]
To assess the activity of Pancreatic Ductal Adenocarcinoma (PDAC) by treatment with CA, gemcitabine (Gem), and doxorubicin (DOX)

• In vitro:Human epithelioid carcinoma attached cell lines Panc-1 and Mia-PaCa-2. Both have increased potential of migration and invasion, as well as Gem resistance

Cytotoxic analysis of CA was measured at 24 and 48 h in combination with Gem and DOX.

CA showed cytotoxic concentrations (IC_50_) of 37.37 µM and 15.06 µM against Panc-1 and Mia-PaCa-2, respectively. Cotreatment with a combination of CA and Gem or DOX did not show synergic activity; in contrast, it showed antagonism, suggesting that CA could display interactions with Gem or DOX.
[135]
To study the effect of CA and DOX on lung cancer

• In vitro: mouse pulmonary system adenocarcinoma LA795 cell line• In vivo: Balb/c mice and Sprague Dawley (SD) rats

In vitro: Not specifiedIn vivo: CA (5.4 mg/kg bw) + DOX (4.1 mg/kg bw)

In vitro: CA inhibited TMEM16A with an IC_50_ of 29.47 ± 3.19 μM. CA regulated the proliferation, migration, and apoptosis of lung cancer cells targeting TMEM16A (binding sites: D439, E448, and R753).CA regulated the growth of lung cancer through the MAPK pathway.CA + DOX inhibited lung cancer cell growth more than a double dose of either one. In vivo: CA + DOX achieved a tumor suppression rate of 85.6% and compensated for side effects.
[132]
To evaluate the effects of tocotrienols and CA encapsulated in a nanoemulsion with cisplatin on lung and liver cancer

• In vivo: human lung cancer cell A549 and liver HEP G2 cancer cells.

Not specified

TRF, CA, and CIS synergistically enhanced late-phase apoptosis and improved cell cycle arrest in the G0/G1 phase.ROS generation was enhanced using TRF:CA:CIS by 16.9% and 30.2% for A549 and HEP G2, respectively.
[129]
To evaluate the oxidative stress induced by multi-walled carbon nanotube (MWCNT) treatment on islets and β-cells.

• In vivo: 
islets and β-cells


CA significantly reduced ROS production after MWCNT treatment and increased insulin secretion together with the enzymes SOD, GSH-Px, CAT, and GSH, but it decreased the level of MDA.
[136]
To evaluate the effect of CA encapsulated in a nanoemulsion on the reduction of nephrotoxicity effects

• In vitro:non-cancer cells of the HEK 293 kidney line

CA (0.08–1.75 μM) + CIS 0.03 μM

Improved cell viability in kidney cells from 33% to over 95%.
[129]
To evaluate delivery systems with CA for the treatment of breast cancer, loaded on oxidant carbon nanotube (OCNT) and/or chitosan (CS).

• In vitro: 
human breast cancer
MDA-MB-231 cell line

CA (100 µg/mL); oxidant carbon nanotube (OCNT)/CA (80 µg/mL); and chitosan (CS)/OCNT/CA (30 µg/mL)

The delivery system based on CS/OCNT/CA showed a higher cytotoxic effect on MDA-MB-231 compared to OCNT/CA and CA alone through apoptosis.
[137]


### 9.2. Clinical Trials

To assess the efficacy of CA in cancer therapy, we searched the clinical trial database clinicaltrials.gov using the keyword “caffeic acid”. One trial with the code NCT02050334, titled “CC100: Safety and Tolerability of Single Doses”, involved 18 participants aged 18 to 65. They were administered CA at a maximum concentration of 24 mg/kg. The findings indicated that CA did not lead to severe adverse effects; the main reported adverse effects were back pain and headache. However, the trial outcomes have not been published.

The effectiveness and safety of CA were assessed in 103 patients with primary immune thrombocytopenia (ITP), with a median age of 48 years. They received an oral regimen of CA tablets at a dose of 300 mg three times per day for 12 weeks. The results demonstrated that CA treatment was effective, with minimal adverse effects, including mild nausea in one case and elevated liver enzymes in another [138]. Furthermore, a meta-analysis involving 2533 patients indicated that CA is statistically effective in treating ITP, leading to an increase in platelet counts (standardized mean difference [SMD] = 1.50, 95% CI [1.09, 1.91], *p* < 0.00001), with few adverse effects (relative risk ratio [RR] = 1.24, 95% CI [1.17, 1.31], *p* < 0.00001) [139]. However, other clinical trials, such as NCT04648917 (GASC1 Inhibitor Caffeic Acid for Squamous Esophageal Cell Cancer [ESCC]), NCT03070262 (The Efficacy and Safety of Caffeic Acid for Esophageal Cancer), and NCT02351622 (Caffeic Acid Tablets as a Second-line Therapy for ITP), have not published their results yet.

## 10. Radioprotective Potential of Caffeic Acid in Chemotherapy

Radiotherapy, a potent cytotoxic treatment, is widely utilized for localized solid tumors, either alone or in conjunction with chemotherapy [140,141]. However, despite its efficacy, it also damages normal cells due to the minimal distinction between cancerous and healthy cells, particularly affecting skin cells, leading to adverse effects such as dermatological conditions and potential cancer development [142]. Notably, nearly 95% of patients undergoing radiotherapy experience symptoms like pain, redness, or ulcers [143]; for instance, pelvic cancer treatment often results in damage to testicular tissue or neuropathy [144,145]. Advanced techniques like intensity-modulated radiotherapy aim to mitigate these toxicities [141]. Nonetheless, akin to chemotherapy, radiotherapy poses significant side effects that can substantially impact the quality of life of patients and treatment adherence [146,147]. Table 4 outlines the protective effects of CA in conjunction with radiotherapy.

## 11. Conclusions

Further research is necessary to comprehensively ascertain the viability of CA as an anticancer treatment. However, preliminary clinical evidence suggests promising benefits. Studies conducted on cells and animals indicate that CA enhances the efficacy of chemotherapy and radiotherapy, potentially mitigating their adverse effects and improving patient outcomes with minimal side effects. This study has some limitations, including a lack of evidence for certain cancer types and omissions in many studies regarding whether the models are drug-resistant or simply aim to potentiate the action in non-resistant cancer cells. Additionally, in vivo testing is limited by small sample sizes, which may restrict the generalizability of the results. Cancer heterogeneity presents another significant limitation. Within a single tumor, cancer cells can vary significantly in differentiation, proliferation rate, metastatic capacity, and therapy susceptibility. This diversity can lead to subpopulations of cells responding differently to specific treatments [151]. Consequently, in vitro models with a single cell type or minimal heterogeneity make it difficult to predict how a tumor with cells in varying degrees of neoplasia will behave. This variability implies that the adjuvant action of CA may differ considerably from patient to patient, even within the same cancer type. While a study may focus on a specific type, the results may not apply to other cases of the same cancer due to biological and molecular differences.

Another interesting property to continue studying is the dual capacity of CA to act as an antioxidant during carcinogenesis and as a pro-oxidant against cancer cells, promoting their apoptosis or sensitizing them to chemotherapeutic drugs. Future challenges include exploring different administration methods for CA as an adjuvant in therapies to treat different neoplasms. Some authors mention that CA’s chemical instability and limited bioavailability restrict its therapeutic potential in vivo. Currently, new formulations are being developed for CA administration in different pathologies, such as transferrin (Tf)-modified nanoparticles (NPs) loaded with CA for Alzheimer’s disease (AD) [152], topical cream with nanostructured lipid carriers (NLCs) loaded with CA for anti-inflammatory action [153], and poloxamer 407 designed to improve the stability of caffeic acid in the stomach [154]. Nonetheless, the utilization of CA in cancer therapy requires more extensive investigation through clinical trials.

## Figures and Tables

**Figure 1 ijms-25-07631-f001:**
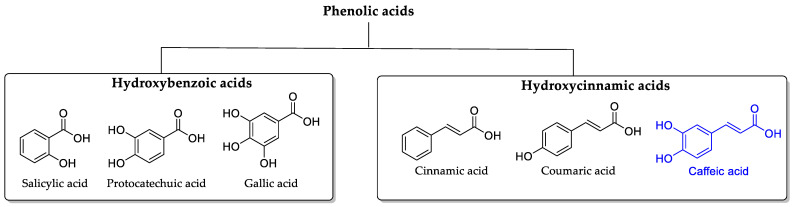
Classification of phenolic acids. Chemical structure of principal hydroxybenzoic and hydroxycinnamic acids, including caffeic acid (CA) in blue (figure created in ChemDraw Ultra 12.0 software).

**Figure 2 ijms-25-07631-f002:**
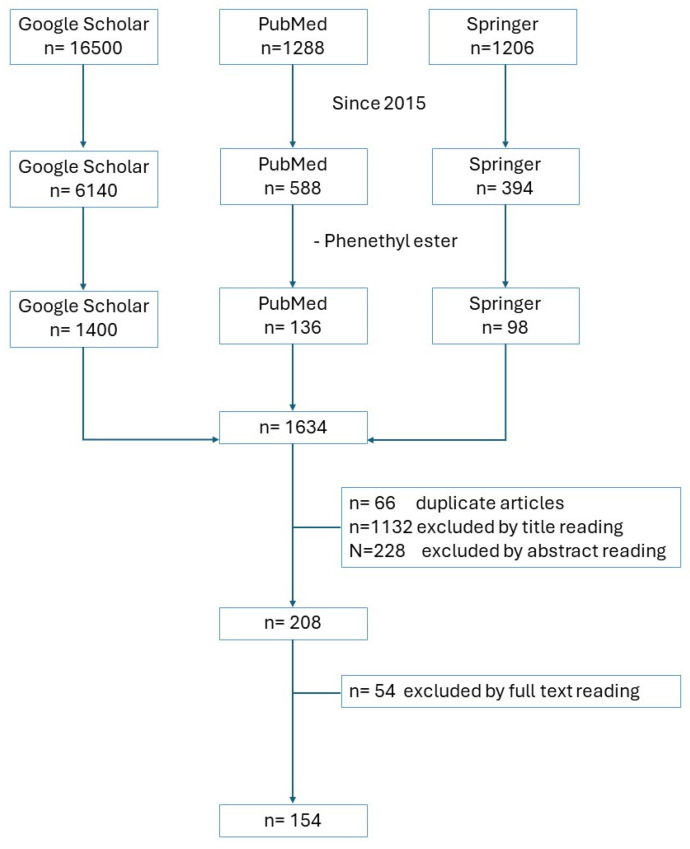
Flowchart with exclusion criteria and number of items found.

**Figure 3 ijms-25-07631-f003:**
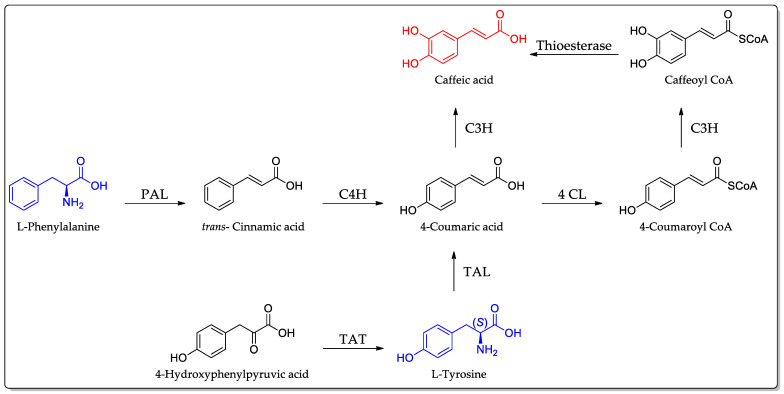
Biosynthetic pathways for the formation of CA (shown in red) from its precursor amino acids (in blue). The involved enzymes are abbreviated: PAL = phenylalanine ammonia-lyase; C4H = cinnamate 4-hydroxylase; 4CL = 4-coumaric acid CoA ligase; TAT = tyrosine aminotransferase; and C3H = *p*-coumarate 3-hydroxylase.

**Figure 4 ijms-25-07631-f004:**
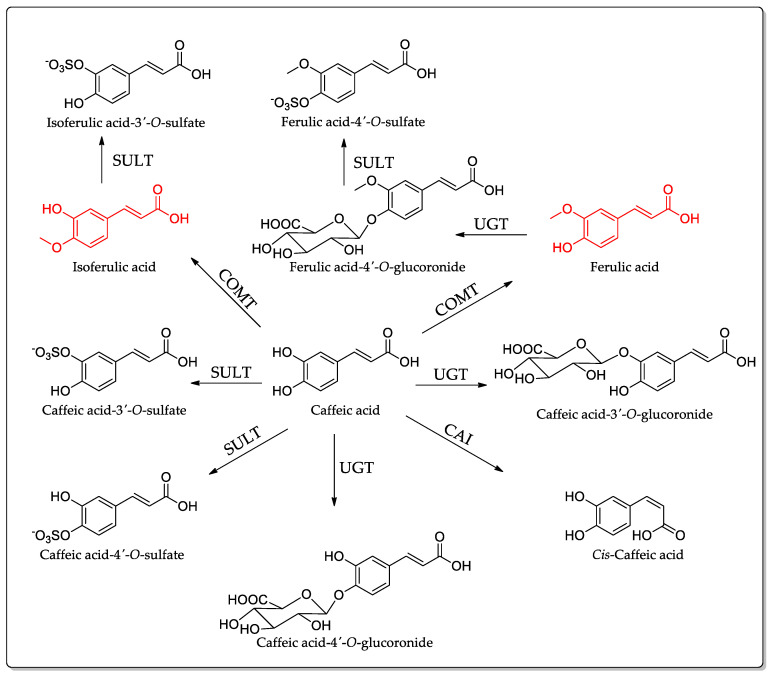
Metabolism of caffeic acid by rats. The involved enzymes are abbreviated: CAI = caffeic acid isomerase; COMT = catechol-*O*-methyltransferase; SULT = sulfotransferase; and UGT = uridine-5′-diphosphate-glucuronosyltransferase [45].

**Figure 5 ijms-25-07631-f005:**
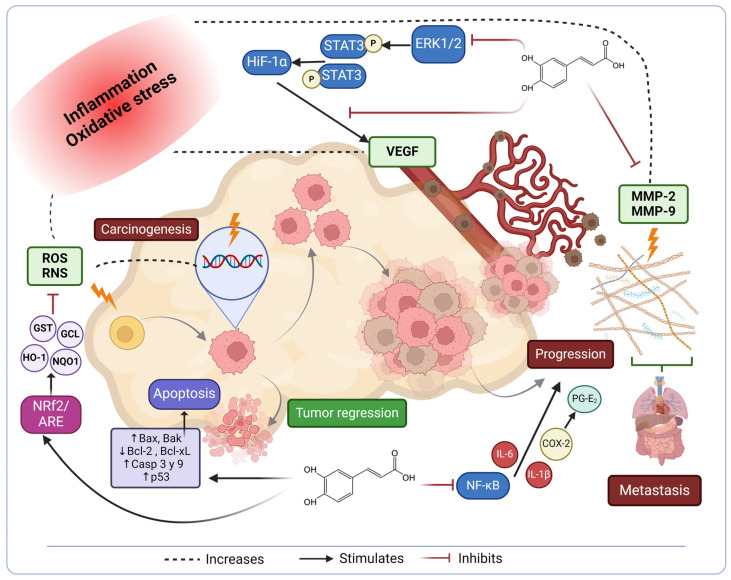
Antioxidant and anti-inflammatory mechanisms modulated by CA in cancer progression.

**Figure 6 ijms-25-07631-f006:**
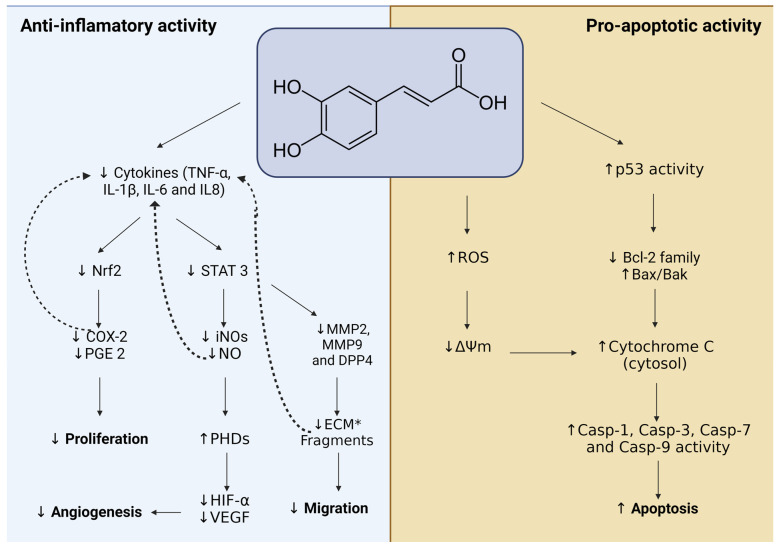
Flowchart summarizing the reported anti-inflammatory and proapoptotic activity of CA in cancer cells. * ECM: extracellular matrix.

**Table 1 ijms-25-07631-t001:** Assessing caffeic acid levels in various plants, including both edible and medicinal plants and plants consumed as is, as used culinary ingredients, or as infusions.

Edible Vegetables or Fruits			Culinary Plants or Herbal Infusion		
Scientific Name/ Common Name	Caffeic Acid mg/kg	References	Scientific Name/ Common Name	Caffeic Acid mg/kg	References
*Allium sativum* (Garlic)	50	[35]	*Andrographis* *paniculata* (Green chiretta)	450	[35]
*Apium graveolens* (Celery)	880–1120	[35]	*Anethum graveolens* (Dill)	1840	[35]
*Averrhoa carambola* (Carambola)	90	[35]	*Artemisia dracunculus* (Tarragon)	620	[35]
*Brassica juncea* (Brown mustard)	830	[35]	*Artemisia pallens* (Davana)	110	[35]
*Capsicum annuum* (Red Pepper)	250–850	[35]	*Artemisia vulgaris* (Mugwort)	25.4	[28]
*Carica papaya* (Papaya)	5080	[35]	*Camellia sinensis* (Tea plant)	510	[35]
*Coriandrum sativum* (Coriander)	240	[35]	*Centella asiatica* (Indian pennywort)	860	[35]
*Eryngium foetidum* (Cilantro)	1600	[35]	*Chromolaena odorata* (Devil weed)	6210	[35]
*Daucus carota* (Carrot)	850	[35]	*Ilex paraguariensis* (Yerba mate)	63.5	[36,37]
*Helianthus annuus* (Sunflower)	30–1100	[35]	*Clerodendrum indicum* (Bharangi)	110	[35]
*Ipomoea aquatica* (Water spinach)	1130	[35]	*Clerodendrum* *thomsoniae*	770	[35]
*Ipomoea batatas* (Sweet Potato)	125	[38]	*Coffea canephora* (Coffea)	12,330	[35]
*Lactuca sativa* (Lettuce)	2580	[35]	*Ginkgo biloba* (Ginkgo)	398	[39]
*Morus alba* (Mulberry)	250	[35]	*Cymbopogon citratus* (Lemon grass)	730	[35]
*Morinda citrifolia* (Noni)	100	[35]	*Echinacea purpurea* (Purple coneflower)	115.9	[28]
*Persea americana* (Avocado)	80	[35]	*Euphorbia hirta* (Asthma plant)	210	[35]
*Persicaria odorata* (Vietnamese coriander)	910	[35]	*Eucommia ulmoides* (Hardy rubber tree)	190	[35]
*Spinacia oleracea* (Spinach)	2.4–5.3 * 370 **	[29] [35]	*Gnaphalium polycaulon* (Western cudweed)	2360	[35]
*Psidium guajava* (Guava)	220	[35]	*Hibiscus sabdariffa* (Roselle)	3510	[35]
*Punica granatum* (Pomegranate)	3050–3630	[35]	*Hyptis suaveolens* (Bamburral)	1110	[35]
*Raphanus sativus* (Radish)	330	[35]	*Leonotis nepetifolia* (Klip dagga)	4180	[35]
*Petroselinum crispum* (Parsley)	480	[35]	*Leonurus sibiricus* (Motherwort)	120	[35]
*Phyllanthus emblica* (Gooseberry)	290	[35]	*Salvia officinalis* (Sage)	1660 ** 74.2 *	[35] [39]
*Physalis angulata* (Poppers)	120	[35]	*Perilla frutescens* (Purple mint)	1890 870	[35] [40]
*Physalis peruviana* (Goldenberry)	860	[35]	*Melissa officinalis* (Lemon balm)	26.8 1740	[35] [28]
*Satureja hortensis* (Summer savory)	248	[41]	*Mentha arvensis* (Pudina)	1080	[35]
*Solanum melongena* (Eggplant)	3.8	[42]	*Mentha cordifolia*	1000	[35]
*Vaccinium myrtillus* (Blueberry)	59.66	[43]	*Mentha piperita* (Peppermint)	57.6	[28]
*Moringa oleifera* (Moringa)	40–300	[35]	*Rosmarinus officinalis* (Rosemary)	1460 ** 43.6 **	[35] [28]
*Thymus vulgaris* (Thyme)	117 * 1550 ** 69.4 **	[39] [35] [28]	*Origanum majorana* (Marjoram)	104 * 67 **	[39] [28]
*Ocimum basilicum* (Basil)	16.6–41.1 * 54.4 ** 3110 **	[29] [28] [35]	*Origanum vulgare* (Oregano)	4100 ** 140 ** 41.4 **	[35] [41] [28]

* mg/kg of fresh weight. ** mg/kg of dried plant.

**Table 2 ijms-25-07631-t002:** Distribution of radioactivity (%) in the gastrointestinal tract of rats after ingestion of [3-^14^C] CA by baggage.

Sample	1 h	3 h	6 h	12 h	24 h	48 h	72 h
Stomach	62	7.1	0.2	0.4	0.1	0.2	0.1
Duodenum	7.9	8.1	0.2	0.3	0.1	0.1	<0.1
Jejunum/ileum	9.1	16	1.1	1.4	1.1	0.7	<0.1
Cecum	<0.1	7.4	6.6	3.9	1.9	0.3	<0.1
Colon	<0.1	7.4	6.6	3.9	1.9	0.3	<0.1

**Table 4 ijms-25-07631-t004:** Protective activity of CA in radiotherapy treatment.

Aim	Model	Treatment Conditions	Finding	Reference
To investigate the radioprotective potential of CA against γ radiation-induced cellular changes	In vitro: human peripheral blood lymphocytes	CA 66 µM for 30 min before γ radiation (1, 2, 3 y, and 4 Gy)	Pre-treatment with CA before γ radiation treatment showed significant cell protection (around 80–85%). Overall, CA protects lymphocytes by decreasing (*p* < 0.01) DNA damage in micronucleus frequencies (MNs) by comet assay, decreasing the level of lipid peroxidation index by TBARS, and improving the antioxidant activity by increasing GSH, SOD, CAT, and GPx levels.	[148]
To investigate the protective role of CA in human epidermal keratinocytes and carcinogenesis induced by cancer treatments with ionizing radiation (γ or X-rays)	In vitro: human epidermal keratinocyte line HaCaT cells	0.1 µg/mL of CA for 24 h prior to γ radiation at 4 Gy (1 Gy/min) for 10 min.	Pre-treatment with CA increased the cell survival significantly (*p* < 0.05) by about 40% at 8 Gy level and reduced ROS production by 38% (*p* < 0.05), which was induced radiation. CA pre-treatment considerably reduced the number of foci of DNA strand breaks at each time point compared to the control.	[149]
To assess the activity of zinc oxide–caffeic acid nanoparticles (ZnO-CA NPs) against cancer cell lines and evaluated on Ehrlich carcinoma treated with γ radiation.	In vitro: human breast cancer MCF-7 cell line and human liver cancer cell line HepG2 In vivo: Ehrlich carcinoma bearing mice (EC mice)	In vitro: Not specified. In vivo: Animals were treated with γ radiation at a dose rate of 0.45 Gy/min in a treatment of 2 Gy/week for 3 successive doses. Animals were injected IP with ZnO-CA NPs (5 mg/100 g) in different experiments.	In vitro: ZnO-CA NPs showed antiproliferative activity against cancer cell lines. The IC_50_ values of ZnO-CA NPs were 9.22 and 11.53 µg/mL for MCF7 and HepG2, respectively. In vivo: ZnO-CA NPs increased the antitumor activity in mice treated with γ radiation. The LD_50_ for ZnO-CA NPs was determined in 50.0 mg/100 g bw. The tumor weight decreased from 56.1% (ZnO-CA NPs) to 71.9% in the combination treatment with γ radiation after 4 weeks compared to untreated solid EC tumor.	[150]

## Data Availability

Not applicable.

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
