# Peer review of "Adjuvant Properties of Caffeic Acid in Cancer Treatment"

_ijms, 2024, doi:10.3390/ijms25147631_

Round 1
Reviewer 1 Report
Comments and Suggestions for Authors
1. Before presenting CA influence on cancer cells, I suggest describing the mechanism of how a cancer cell is formed, with particular reference to various pathways, e.g. oxidative and inflammatory, ErbB family pathway, the p53-mediated apoptosis pathway, etc. This should be presented as a separate chapter. Then the publication will be more readable and the reader will have a latitude to relate certain facts
2. Section 8 should be divided into two demonstrating both the in vitro and in vivo (animals) and in vivo (clinical) studies. The Authors provide with a separate chapter 10 with clinical data, however, in my opinion, this should be inserted as a subsection of the chapter 8. otherwise, it is quite chaotic
3. Also, the paper would be improved greatly if the authors could provide information on whether CA is effective on every cancer cell types. What types of cancer are resistant?
4. Please provide the limitation of the study
2.
Comments on the Quality of English Languageminor changes are required
Author Response
Dear Reviewer.
We would like to thank your comments and suggestions done to our ms, we received some very similar comments from other reviewers, then we tried to do our best to give a better response to all of you. You can see that a new paragraph was added, tables, and an extensive number of citations. I hope that this new version is to your agree.
We give response point to point of your comments.
All the best!
Dr. Cristian Paz
Reviwer 1.
(x) Minor editing of English language required
- Before presenting CA influence on cancer cells, I suggest describing the mechanism of how a cancer cell is formed, with particular reference to various pathways, e.g. oxidative and inflammatory, ErbB family pathway, the p53-mediated apoptosis pathway, etc. This should be presented as a separate chapter. Then the publication will be more readable and the reader will have a latitude to relate certain facts
Response: Chapter 6 about Signaling pathways affected in cancer progression was added before presenting CA influence on cancer cells.
- Section 8 should be divided into two demonstrating both the in vitro and in vivo (animals) and in vivo (clinical) studies. The Authors provide with a separate chapter 10 with clinical data, however, in my opinion, this should be inserted as a subsection of the chapter 8. Otherwise, it is quite chaotic
Response: Your suggestion was accepted, and the chapter was divided into 2 sections, one dealing with in vitro and in vivo assays, and another section dealing with clinical assays. In addition, we added an extra column with the specific model to Tables 3 and 4 for the reader's clarity.
- Also, the paper would be improved greatly if the authors could provide information on whether CA is effective on every cancer cell types. What types of cancer are resistant?
Response: Resistant Cell lines have been mentioned. It is effective in some cancers, but has not yet been tested in all cancers, so this is an issue we discuss in the conclusions as a limitation of the study.
- Please provide the limitation of the study
Response: Limitations of the study were added in the Conclusions section.
Reviewer 2 Report
Comments and Suggestions for Authors
In this review authors screened the current literaure from Google Scholar, PubMed, the Phenol-Explorer database, and ClinicalTrials.gov to underscore the potential of CA in cancer prevention and overcoming chemoresistance.
The manuscript is interesting and generally well written and illustrated. However, several points deserve to be improved. My comments are listed below.
Figure 1: Where these structures come from? did the authors took them from a free database? the source must be specified
Line 60: Since this is a review article, authors must give a general overview of the topic treated. In particular, since oxidative stress plays a key role in this manuscript, a brief introduction on NRF2/KEAP1/ARE signaling deserves to be added. In fact, this pathway plays a key role in the onset and progression of several cancer types (see PMID: 37525922 and PMID: 37841775 ).
2. Methodology: A flow chart reporting inclusion and exclusion criteria, as well as the number of studies obtained by each database screened deserves to be added
Figure 2: Why L-Tyrosine and 4-Hydroxyphenylpiruvate have been reported in blue?
Table 3 and 4: it should be added a column reporting the model used in the study discussed
A schematic figure resuming the main topics discussed in this review should be added
A graphical abstract would be helpful for the readers interested in this topic
An accurate revision of typing errors is recommended.
An accurate revision of formatting is recommended. In particular, the presence of short paragraph and spaces among paragraphs should be avoided.
Abbreviations must be written in full length when mentioned ofr the first time
Author Response
Dear Reviewer.
We would like to thank your comments and suggestions done to our ms, we received some very similar comments from other reviewers, then we tried to do our best to give a better response to all of you. You can see that a new paragraph was added, tables, and an extensive number of citations. I hope that this new version is to your agree.
We give response point to point of your comments.
All the best!
Dr. Cristian Paz
Reviewer 2
In this review authors screened the current literaure from Google Scholar, PubMed, the Phenol-Explorer database, and ClinicalTrials.gov to underscore the potential of CA in cancer prevention and overcoming chemoresistance.
The manuscript is interesting and generally well-written and illustrated. However, several points deserve to be improved. My comments are listed below.
Figure 1: Where these structures come from? did the authors took them from a free database? the source must be specified
Response: The source was specified. It made it by the authors using ChemDraw software.
Line 60: Since this is a review article, authors must give a general overview of the topic treated. In particular, since oxidative stress plays a key role in this manuscript, a brief introduction on NRF2/KEAP1/ARE signaling deserves to be added. In fact, this pathway plays a key role in the onset and progression of several cancer types (see PMID: 37525922 and PMID: 37841775 ).
Response: Information on the role of oxidative stress and the NRF2/KEAP1/ARE pathway has been added in a new chapter 6 on Signaling pathways affected in cancer progression.
- Methodology: A flow chart reporting inclusion and exclusion criteria, as well as the number of studies obtained by each database screened deserves to be added
Response: A flow chart was added in Methodology section.
Figure 2: Why L-Tyrosine and 4-Hydroxyphenylpiruvate have been reported in blue?
Response: It was corrected. L-Tyrosine and L-Phenilalanine were reported in blue to highlight the amino acids involved in caffeic acid biosynthesis.
Table 3 and 4: it should be added a column reporting the model used in the study discussed
Response: We added an extra column with the specific model to Tables 3 and 4 for the reader's clarity.
A schematic figure resuming the main topics discussed in this review should be added
Response: Flowchart summarizing the activity of CA in cancer cells was added as Figure 6, Chapter 8.
A graphical abstract would be helpful for readers interested in this topic
Response: A graphical abstract was added
An accurate revision of typing errors is recommended.
Response: It was reviewed and corrected
An accurate revision of formatting is recommended. In particular, the presence of short paragraphs and spaces among paragraphs should be avoided.
Response: It was reviewed and corrected
Abbreviations must be written in full length when mentioned for the first time
Response: It was revised and corrected.
Round 2
Reviewer 2 Report
Comments and Suggestions for Authors
the manuscript can be accepted in the present form